# Evolutionarily Developed Alternatively Spliced Exons Containing Translation Initiation Sites

**DOI:** 10.3390/cells14010011

**Published:** 2024-12-26

**Authors:** Jun-ichi Takeda, Takaaki Okamoto, Akio Masuda

**Affiliations:** 1Center for One Medicine Innovative Translational Research (COMIT), Institute for Advanced Study, Gifu University, 1-1 Yanagido, Gifu 501-1193, Japan; takeda.junichi.a3@f.gifu-u.ac.jp; 2Center for Neurological Diseases and Cancer, Nagoya University Graduate School of Medicine, 65 Tsurumai, Showa-ku, Nagoya 466-8550, Japan; okamoto.takaaki.a2@f.mail.nagoya-u.ac.jp; 3Academia-Industry Collaboration Platform for Cultivating Medical AI Leaders (AI-MAILs), Nagoya University Graduate School of Medicine, 65 Tsurumai, Showa-ku, Nagoya 466-8550, Japan

**Keywords:** alternative splicing, large exon, vertebrate evolution, MATR3, autoregulation

## Abstract

Alternative splicing is essential for the generation of various protein isoforms that are involved in cell differentiation and tissue development. In addition to internal coding exons, alternative splicing affects the exons with translation initiation codons; however, little is known about these exons. Here, we performed a systematic classification of human alternative exons using coding information. The analysis showed that more than 5% of cassette exons contain translation initiation codons (alternatively skipped exons harboring a 5′ untranslated region and coding region, 5UC-ASEs) although their skipping causes the deletion of translation initiation sites essential for protein synthesis. The splicing of 5UC-ASEs is under the repressive control of MATR3, a DNA/RNA-binding protein associated with neurodegeneration, and is distinctly regulated particularly in the human brain, muscle, and testis. Interestingly, MATR3 represses its own translation by skipping a 5UC-ASE in *MATR3* to autoregulate its expression level. 5UC-ASEs are larger than other types of alternative exons. Furthermore, evolutionary analysis revealed that 5UC-ASEs have already appeared in cartilaginous fishes, have increased in amphibians, and are concentrated in the genes involved in transcription in mammals. Taken together, our analysis identified a unique set of alternative exons, 5UC-ASEs, that have evolutionarily acquired a repression mechanism for gene expression through association with MATR3.

## 1. Introduction

The development of sequencing technology has enabled us to view the landscape of the human genome and transcriptome [1,2,3]. There are ∼19,000 protein-coding genes and >20,000 non-coding RNA genes, containing ~300,000 exons, in the human genome [4,5]. More than 50% of these exons are alternatively spliced, 40% of which are in protein-coding regions [5]. Alternative splicing generates multiple transcript isoforms from a single gene, increasing the diversity of the human transcriptome and proteome.

The dynamic regulation of alternative splicing plays a pivotal role in tissue development, particularly in the brain and striated muscle [6]. The dynamics of these alternative splicing events, which vary in different tissues and are conserved among mammals [5,7], orchestrate the expression of protein isoforms to promote cell differentiation. Their perturbations cause tissue-specific degenerative diseases in humans [6].

Splicing is facilitated by the spliceosome composed of splicing trans factors, including five small nuclear ribonucleoproteins (snRNPs), namely U1, U2, U4, U5, and U6. The U snRNPs recognize their cognate *cis*-elements, such as the 5′ and 3′ splice sites, the polypyrimidine tract, and the branch point, to drive splicing. The alternative exons are typically characterized by weak splice sites, which mitigates their recognition by the spliceosome [8]. Instead, the *cis*-elements called exonic/intronic enhancer/silencer elements (ESE, ISE, ESS, and ISS, respectively) are enriched within and around alternative exons. RNA-binding proteins (RBPs), such as serine/arginine-rich splicing factors (SRSFs) and heterogeneous nuclear ribonucleoproteins (hnRNPs), are recruited to these elements to modulate spliceosome activity [9]. In addition, RBPs expressed in specific tissues, such as RBFOX and NOVA, are recruited to their cognate elements to achieve the tissue-specific regulation of alternative splicing [10,11,12].

The sizes of internal exons in humans typically range from 50 to 250 nucleotides, suggesting the existence of an optimal exon length for efficient splicing [13]. Alternative exons are usually smaller than constitutive exons (122 nt vs. 136 nt, mean) [14]. Microexons (<30 nt) are specifically spliced in neuronal cells with the help of the neuron-specific RBP, SRRM4 [15]. In contrast, large exons (>300 nt) are spliced in a wide range of tissues with the help of the ubiquitously expressed RBP, SRSF3 [16]. Although most of alternative exons show low evolutionary conservation, even in primates, the microexons and the large exons show deep evolutionary conservation across vertebrates [15,16,17]. These findings suggest the existence of different classes of alternative exons whose splicings are under the control of specific RBPs; however, this remains to be elucidated.

MATR3 is a DNA/RNA-binding protein associated with neuromuscular degenerative diseases such as amyotrophic lateral sclerosis, frontotemporal lobar degeneration, and distal myopathy [18,19]. MATR3 preferentially binds to U-nucleotide-enriched regions in introns of pre-mRNA to suppress the splicing of nearby cassette exons [20]. In addition, MATR3 is implicated in DNA-regulating activities including chromosomal organization and transcription [21]. MATR3 expression levels vary among different tissues and are abundant in developing neuronal tissues [22,23] although the regulatory mechanism for MATR3 expression remains unelucidated.

Here, we classify ~300,000 human internal exons based on coding information. We find that the exons containing translation initiation sites are larger and more frequently skipped than those within coding sequences. These exons with translation initiation sites are alternatively spliced, particularly in the brain, muscle, and testis. In cellulo analysis revealed that MATR3 specifically binds to the upstream introns of the alternatively skipped exons harboring a 5′ untranslated region and coding region (5UC-ASEs) to repress their splicing. Evolutionary analysis reveals that 5UC-ASEs have already appeared in cartilaginous fishes, have increased in amphibians, and are concentrated in the genes involved in transcriptional regulation in mammals. In summary, we identified a unique class of alternative exons, 5UC-ASEs, which had not been identified in the previous analyses based on sequence composition [24,25].

## 2. Materials and Methods

### 2.1. Phylogenetic Classification of Human Exons Using In Silico Analyses

#### 2.1.1. Classification of Human Exons Based on Coding Information

Protein-coding genes were extracted according to the human basic gene annotation data (gencode.v43.basic.annotation.gtf) and UniProtKB/SwissProt metadata (gencode.v43.metadata.SwissProt) of GENCODE Release 43 [26]. A unique internal exon with no other overlapping transcript isoforms was defined as a constitutively spliced exon. When multiple transcript isoforms with internal exons were generated from a gene, an internal exon present in all transcript isoforms was also defined as a constitutive exon. The other exons were classified as alternative exons. The alternative exons were further subdivided into cassette exons (alternatively skipped exons, ASEs), exons with alternative 5′ or 3′ splice sites (splice-site-shifted exons, SSEs), and the other alternative exons. An internal exon present in at least one but not all of the transcript isoforms covering the exon region was defined as an ASE. A cluster of overlapping exons with different 5′ or 3′ splice sites was defined as an SSE.

#### 2.1.2. Estimation of the Tissue-Specific PSIs of Human 5UC-ASEs

To detect skipping exon events of human 5UC-ASEs, gencode.v43.basic.annotation.gtf was analyzed using SUPPA version 2.3 [27] with the parameter “generateEvents -f ioe -e SE”. Percent spliced-in (PSI) values [28] of the detected 1001 human 5UC skipping exon events were calculated based on “Exon-exon junction read counts” (GTEx_Analysis_2017-06-05_v8_STARv2.5.3a_junctions.gct) in Genotype-Tissue Expression (GTEx) Analysis V8 [29].

The following formula was used to calculate PSI with the threshold of total read counts > 50: (upstream junction reads + downstream reads)/(upstream junction reads + downstream reads + skipping junction reads × 2).

PSI values in thirty human tissues annotated in “A de-identified, open access version of the sample annotations in dbGaP” (GTEx_Analysis_v8_Annotations_SampleAttributesDS.txt) were estimated.

#### 2.1.3. Detection of 5UC-ASEs and Genes Orthologous to the Human Genes That Contain 5UC-ASEs in Model Organisms

5UC-ASEs in nine vertebrates were detected using the same method employed for the detection of human 5UC-ASEs. The gene annotation data of rhesus macaque (Macaca_mulatta.Mmul_10.105.chr.gtf), mouse (Mus_musculus.GRCm39.105.chr.gtf), rat (Rattus_norvegicus.mRatBN7.2.105.chr.gtf), cow (Bos_taurus.ARS-UCD1.3.111.chr.gtf), chicken (Gallus_gallus.bGalGal1.mat.broiler.GRCg7b.107.chr.gtf), frog (Xenopus_tropicalis.UCB_Xtro_10.0.107.chr.gtf), zebrafish (Danio_rerio.GRCz11.105.chr.gtf), medaka (also known as Japanese rice fish, Oryzias_latipes.ASM223467v1.105.chr.gtf), and elephant shark (Callorhinchus_milii.Callorhinchus_milii-6.1.3.105.gtf) were downloaded from Ensembl [30]. Orthologous genes were extracted using “GET homology/id/:species/:id?type=orthologues;format=condensed” of Ensembl REST API version 15.7 [31]. The orthologous comparison of 5UC-ASE genes among ten organisms including humans was performed by round-robin to create a dendrogram as phylogenetic tree. The hierarchical clustering and dendrogram plotting were run with *R* version 4.2.3. The dendrogram was constructed using Ward’s algorithm, a commonly used method for creating phylogenetic trees.

#### 2.1.4. Gene Ontology (GO) Analysis of the Genes That Contain 5UC-ASEs

GO analysis of the genes with 5UC-ASEs in each species was conducted using the Database for Annotation, Visualization, and Integrated Discovery (DAVID) [32,33]. GO terms associated with at least 5% of the input genes were extracted from the results of DAVID’s GO BP DIRECT and indicated. In Figure 1D, g:Profiler [34] was used for the GO analysis of the genes containing 5UC exons (6193 genes) because of the limitation of DAVID, which can analyze less than 3000 genes.

The annotations for lizard (Anolis_carolinensis.AnoCar2.0v2.105.chr.gtf), sea lamprey (Petromyzon_marinus.Pmarinus_7.0.105.gtf), and sea squirt (Ciona_intestinalis.KH.106.chr.gtf) contain much fewer protein-coding genes than the above species, making it difficult to estimate the exact number of 5UC-ASEs. Therefore, we excluded these species from the analysis.

### 2.2. In Silico Analysis of Functional RBPs Binding Around 5UC-ASEs

To investigate the bindings of splicing factors to RNA around 5UC-ASEs, we analyzed 42 and 47 eCLIP datasets of HepG2 and K562 cells, respectively, in ENCODE [35] for 33 RBPs associated with the GO term “splicing” in AmiGO2 [36,37,38] (Appendix A). eCLIP shows the transcriptome-wide distributions of in cellulo binding sites of an RBP of interest. We limited our analysis to 1398 5UC-ASEs in the Swiss-Prot-validated coding genes to investigate the involvement of 5UC-ASEs in translational regulation (Appendix A). At first, eCLIP bam format files aligned to GRCh38 of these RBPs were converted into bigWig format files using deepTools version 3.5.2 with bamCoverage --binSize 10 [39]. Then, coverage scores per genome region were calculated using deepTools with computeMatrix --beforeRegionStartLength 500 --regionBodyLength 100 --afterRegionStartLength 500 –skipZeros to reveal the binding of an RBP from −500 bp upstream to +500 bp downstream around the 5UC-ASEs. The same processing was also executed against 14,832 Type 3 ASEs and 341 Type 4 ASEs. To depict the volcano plots, we calculated the median fold-change of coverage scores between 5UC-ASEs (Type 2) and Type 3 ASEs in each region and their *p*-values by exact Wilcoxon signed rank test using *R*. To quantify the PSIs of 5UC-ASEs and Type 4 ASEs in *MATR3*/*TIA1*-knocked-down HepG2/K562 cells, the transcript quantification files (tsv files) of shRNA-seq and control RNA-seq registered in ENCODE (Appendix A) were analyzed. Similar to the estimation of the tissue-specific PSIs of human 5UC-ASEs (2.1.2), skipping exon events in the coding genes were generated from “Basic gene annotation” (gencode.v43.basic.annotation.gtf) using SUPPA version 2.3 with generateEvents -f ioe -e SE, and 5UC-ASEs, Type 3 ASEs, and Type 4 ASEs were extracted from them. Then, the PSI values of these exons were estimated based on the transcripts per million (TPM) values described in the transcript quantification files using SUPPA with psiPerEvent. The difference between PSIs [shRNA–control RNA: Delta PSI (ΔPSI)] was calculated by SUPPA with diffSplice --method empirical -gc. The coverage scores were calculated by deepTools with computeMatrix --beforeRegionStartLength 500 --regionBodyLength 100 --afterRegionStartLength 500 –skipZeros using the converted bigWig format files of MATR3 and TIA1 eCLIP datasets of HepG2 and K562 cells to analyze the RBPs binding around 5UC-ASEs, Type 3 ASEs, and Type 4 ASEs. The statistical differences in the coverage scores among the three categories were tested by Steel–Dwass test with source (“http://aoki2.si.gunma-u.ac.jp/R/src/Steel-Dwass.R (accessed on 5 May 2024)”, encoding=“euc-jp”) on *R*.

### 2.3. Antibodies

Anti-N-terminal MATR3 antibody (ARP40922_T100) was purchased from Aviva Systems Biology, CA, USA. Anti-C-terminal MATR3 antibody (ab151714) [22] was purchased from Abcam, Cambridge, United Kingdom. Anti-GAPDH antibody (G9545) was purchased from Sigma-Aldrich, MO, USA. Anti-GFP antibody (11814460001) was purchased from Roche, Basel, Switzerland.

### 2.4. Cell Cultures and Transfection

HEK293 (RCB1637), HeLa (RCB0007), and Jurkat (RCB0537) cells were provided by the RIKEN BioResource Research Center (BRC), Ibaraki, Japan. These cells were grown in DMEM with 10% fetal bovine serum at 37 °C in 5% CO_2_. HEK293 cells were transfected with plasmids using Lipofectamine 3000 (Thermo Fisher Scientific, Waltham, MA, USA) according to the manufacturer’s instructions.

### 2.5. RNA Extraction and RT-PCR

Total RNA was isolated at 48 h after transfection using RNeasy Mini Kit (QIAGEN, Hilden, Germany) according to the manufacturer’s instructions, then reverse-transcribed to make first-strand cDNA using an oligo-dT primer (Thermo Fisher Scientific) and ReverTraAce (TOYOBO, Osaka, Japan). PCR was performed with GoTaq polymerase (Promega, WI, USA) using the following primer set.

For the analysis of endogenous MATR3 mRNA, forward primer was 5′-GGGGGATTGTGGGAGTCTCC-3′; reverse primer was 5′-TCTTGGTCCAACTGCTGGTC-3′.

For the analysis of pSPL3 minigene mRNA, forward primer was 5′-TCTGAGTCACCTGGACAACC-3′; reverse primer was 5′-ATCTCAGTGGTATTTGTGAGC-3′.

Total RNAs extracted from human tissues were purchased from Clontech (smooth muscle, #636547; skeletal muscle, #636534; heart, #636532; brain, #636530; cortex, #636561; liver, #636544; spleen, #636525).

### 2.6. Immunoblotting

Cells were collected by centrifugation at 2000× *g* for 1 min and suspended in NETN buffer (20 mM Tris-HCl pH 8.0, 1 mM EDTA, 120 mM NaCl, and 0.5% NP-40) with protease inhibitors (1 μg/μL aprotinin, 1 μg/μL leupeptin, and 1 mM PMSF). Following sonication for 30 s with an ultrasonic disruptor (UR-20P, Tomy Seiko, Tokyo, Japan), each sample was centrifuged, and the supernatant was harvested as total cell lysates. SDS-PAGE and Western blotting were performed as previously described [16] with the antibodies indicated in the figures.

### 2.7. Polysome Profiling Analysis

HEK293 cells were collected by centrifugation at 2000× *g* for 1 min and suspended in cytoplasmic lysis buffer (10 mM HEPES-KOH pH 7.8, 10 mM KCl, 0.1 mM EDTA, 0.1% NP-40, and 1 mM 1,4-dithiothreitol) with cOmplete proteinase inhibitor cocktail (Roche) and RNase inhibitor (TOYOBO). The solution was centrifuged at 5000× *g* for 2 min and the supernatant was harvested as cytoplasmic lysates. The cytoplasmic lysates were loaded onto a 15–45% sucrose gradient prepared using a Gradient Master 108 (BioComp Instruments, Fredericton, NB, Canada) and centrifuged at 36,000× *g* in a Beckman SW 41 Ti rotor for 150 min. Then, 24 fractions were collected, tracing optical density at 260 nm using a Piston Gradient Fractionator (BioComp Instruments). RNA was isolated from each fraction and RT-PCR was performed as described above.

### 2.8. Construction of Minigenes and Expression Vectors for Splicing Analysis

MATR3 minigene (pSPL3-MATR3_ex2) was constructed by amplifying exon 2 and flanking intronic regions of human *MATR3* (851 nucleotides of intron 1 and 1353 nucleotides of intron 2) using a proofreading DNA polymerase (PrimeSTAR, Takara, Shiga, Japan). The amplified fragment was cloned into the modified splicing reporter vector, pSPL3 [40], at NotI and PacI sites. Human *MATR3* (accession number, BC015031) CDS was purchased from Open Biosystems and was cloned into a p3xFLAG-CMV10 mammalian expression vector at NotI and KpnI sites. The CMV-expression vector encoding MATR3 fused to EGFP (MATR3-EGFP) was constructed by the cloning of human *MATR3* CDS with 3xFLAG CDS into a pEGFPN1 vector at EcoRI and BamHI sites.

### 2.9. Tethered Function Assay of MATR3

To make a *MATR3* minigene carrying 24xMS2 hairpin loops (pSPL3-MATR3_ex2+24xMS2), the 24xMS2 hairpin-loop sequence was PCR-amplified using CFP_ betaGlobin_24XPP7_24XMS2_Splicing_ reporter [41], a gift from Daniel Larson (Addgene plasmid # 61762), as a template. The purified 24xMS2 PCR product was introduced at the PacI site using In-Fusion HD cloning kit (Takara). To make mammalian vectors expressing the MATR3-MS2 proteins, the coding sequence of MS2-coat protein was amplified from a vector carrying the MS2-hnRNP H fusion cDNA [42] and was inserted into the 3′ ends of coding regions of MATR3-EGFP in pEF-Bos-MATR3-EGFP [43]. As controls, the expression vector of MS2 coat protein alone (MS2), that of MS2 fused with EGFP (MS2-EGFP) [42], and that of MATR3 fused with EGFP (MATR3-EGFP) were prepared. Artificial tethering of MATR3 was performed by co-transfection of a MATR3 minigene carrying 24xMS2 hairpins and an effector construct with or without MS2 coat protein.

## 3. Results

### 3.1. In Silico Analysis Identified a Unique Class of Exons Containing 5′ Untranslated Region (UTR) and Coding Sequence (CDS)

#### 3.1.1. The Classification of Human Internal Exons Based on the Coding Information

In order to identify a new class of exons with unique sequence features, we classified all the internal exons in the human coding genes according to the coding information. We analyzed the GENCODE Release 43 human gene annotation information and extracted 141,133 unique internal exons in 12,754 multi-exon coding genes. We divided the internal exons into five categories based on the existence of coding and non-coding regions (Figure 1A, top panel, and Table 1) as follows: exons completely within 5′ UTR (Type 1 exons, 5651 exons), exons consisting of a 5′ UTR and CDS region (Type 2 [5UC] exons, 6610 exons), exons completely within the CDS region (Type 3 exons, 127,144 exons), exons consisting of a CDS region and 3′ UTR (Type 4 exons, 1556 exons), and exons completely within 3′ UTR (Type 5 exons, 172 exons). We observed that most of the internal exons were distributed within CDS regions (Type 3 exons, 127,144 exons) as predicted. In addition, substantial and small numbers of them contained 5′ UTRs (Type 1 and Type 2 exons, 12,261 exons) and 3′ UTRs (Type 4 and Type 5 exons, 1728 exons).

We then divided these internal exons into constitutive and alternative exons by referring to the transcript annotation information in gencode.v43.basic.annotation.gtf to understand how these classes of exons are differentially spliced. The alternative exons were further subdivided into cassette exons (alternatively skipped exons, ASEs), those with alternative 5′ or 3′ splice sites (splice-site-shifted exons, SSEs), and the other alternative exons, most of which were generated by the alternative splicing within terminal exons. We observed that Type 3 exons predominantly contained constitutive exons, presumably to the avoid disruption of their coding sequences (Figure 1A). In contrast, 5UC (Type 2) and Type 3 exons predominantly contained alternative exons despite the presence of coding sequences, implying the unique role of these exons in their gene expressions.

We estimated the individual lengths of these exons as the classes of internal exons with distinct exon lengths had been reported [16]. The median length of all the internal exons was 122 nt, which was consistent with a previous report [13]. Our analysis revealed that the median length of 5UC (Type 2) exons was remarkably large (161 nt) while those of the other exon categories were around 122 nt (Figure 1B). Large exon sizes were similarly observed for constitutively spliced and alternatively spliced 5UC exons (Appendix A).

We next analyzed the codon frequency of 5UC exons since we had previously identified a set of large exons enriched in the codes for proline and serine [16]. However, 5UC (Type 2) exons did not show such codon bias when compared to Type 3 and Type 4 exons (Figure 1C). Only the codon for methionine was prominent, reflecting the presence of translation initiation sites. Gene ontology analysis showed a significant association of 5UC exons with DNA binding and transcriptional regulation (Figure 1D). These results suggest that 5UC exons represent a distinct class of internal exons.

#### 3.1.2. Alternative Splicing of 5UC Exons in Human Tissues

We observed that a substantial number of the 5UC exons are alternatively skipped (5UC-ASEs, 1247 exons) although their skipping result in the absence of translation initiation codons essential for protein synthesis. We confirmed that 5UC-ASEs were concentrated in the genes involved in DNA binding and transcriptional regulation and that they did not show remarkable codon bias except for methionine (Appendix A), as was observed for 5UC exons (Figure 1C). To investigate the actual splicing status of 5UC-ASEs in human tissues, we analyzed their PSI values in 30 human tissues using RNA-seq data of the Genotype-Tissue Expression (GTEx) Consortium [44]. We observed that the PSI values showed bimodal distributions with peaks near 0 and 1.0 in all the 30 tissues. Alternative splicing with moderate PSIs (0.3 < PSI < 0.75) was observed in ~15% of 5UC-ASEs in most tissues although the percentages were elevated especially in the brain, muscle, and testis (>20%) (Figure 2A). We then estimated how the splicing of an exon is altered in a given tissue by calculating the difference of the exon’s PSI in the tissue from the median of them in all tissues (tissue-ΔPSI). We found that the absolute values of tissue-ΔPSI were significantly high in the brain, muscle, and testis among the 30 tissues analyzed (Figure 2B). These results suggest that the alternative splicing of 5UC-ASEs is distinctly regulated in these three tissues.

#### 3.1.3. Development of 5UC-ASEs During Vertebrate Evolution

We next analyzed the evolution of 5UC-ASEs in vertebrates. By investigating the genomic information as was performed in the analysis of human 5UC-ASEs, we extracted 5UC-ASEs in vertebrate species including the rhesus macaque from Primates, mouse and rat from Rodentia, cow from Artiodactyla (above are mammalia), chicken from Aves, frog from Amphibia, zebrafish and medaka from Actinopterygii, and elephant shark from Chimaeriformes (Figure 2C and Table 2). Our analysis showed that 5UC-ASEs had appeared at least in Chimaeriformes and had markedly increased their presence in frogs, suggesting that 5UC-ASEs had become prevalent in Amphibia. We also performed a phylogenetic evolutionary analysis of genes containing 5UC-ASEs (5UC-ASE genes) based on the amino acid sequences and found that they can be classified into several groups (Figure 2D), which was similar to the actual phylogenetic tree created by TimeTree (https://timetree.org/ accessed on 11 October 2024) inputting their corresponding NCBI taxonomy names [45]. In addition, GO analysis showed that 5UC-ASEs genes had been concentrated in transcriptional regulatory genes from mammals (Figure 2E and Appendix A). These results suggest that 5UC-ASEs have acquired a role in transcriptional regulation during vertebrate evolution.

### 3.2. Identification of MATR3 as a Splicing Repressor of 5UC-ASEs

Our analysis demonstrated the tissue-specific alternative splicing of 5UC-ASEs (Figure 2A,B). To identify splicing factors responsible for the splicing regulation, we dissected the eCLIP database in ENCODE. eCLIP reveals transcriptome-wide in vivo RNA sites bound by an RBP of interest. We analyzed the eCLIP read distributions for 33 splicing factors in HepG2 and K562, from 500 nt upstream to 500 nt downstream, of 5UC-ASEs (Type 2-ASE) and Type 3 ASEs (Figure 3A: an example of MATR3 eCLIP analysis). For comparison, we also analyzed the read distributions around Type 4 ASEs (Appendix A). We estimated the fold-change and *p*-value of read coverage between 5UC-ASEs and Type 3 ASEs in upstream intronic regions, exons, and downstream intronic regions. Volcano plots showed that MATR3 and TIA1 bindings had accumulated in the upstream regions of 5UC-ASEs consistently in both cells (cut-off: log_2_ FC > 0.4 and log_10_ *p*-value > 6. Figure 3B). MATR3 accumulation was particularly evident in the upstream intronic regions of 5UC-ASEs (Figure 3A) although it had also accumulated around Type 4 ASEs (Appendix A). We then performed an integrated analysis of MATR3/TIA1-eCLIP and RNA-seq of *MATR3*/*TIA1* silenced cells to understand how these bindings affect the alternative splicing of 5UC-ASEs. We classified 5UC-ASEs into three categories based on the ΔPSI values by *MATR3*/*TIA* silencing (ΔPSI ≥ 0.2, −0.2 < ΔPSI < 0.2, and ΔPSI ≤ −0.2) and found that MATR3, but not TIA1, is significantly clustered on upstream intronic regions of the 5UC-ASEs included by the silencing in both HepG2 and K562 cells (Figure 3C). These results suggest that the binding of MATR3 upstream to 5UC-ASEs promotes the skipping of 5UC-ASEs.

### 3.3. Abundant Expression of the MATR3 mRNA Isoform Lacking a 5UC-ASE, Exon 2

MATR3 plays diverse roles in transcription and RNA processing. In particular, MATR3 is known to function as a splicing repressor that preferentially binds intronic regions [21]. Interestingly, the *MATR3* gene contains a 5UC-ASE, exon 2, which is 1089 nt long, consistent with our observation that 5UC exons are generally large (Figure 1B). RT-PCR analysis confirmed that *MATR3* exon 2 is alternatively skipped in various human tissues and cell lines (Figure 4A). The skipping is notable in neuronal tissues in combination with the skipping of exon 3. In addition, the overexpression of EGFP-tagged MATR3 (MATR3-EGFP) promoted the skipping of exon 2 in endogenous *MATR3* mRNA in HEK293 cells (Appendix A), representing the repressive role of MATR3 for the splicing of 5UC-ASEs (Figure 3C).

Our CLIP-seq of HEK293 cells (DDBJ: DRR494386) [43], as well as eCLIP datasets of K562 and HepG2 cells, showed marked enrichments of MATR3 bindings around exon 2 (Figure 4B and Appendix A). More bindings were observed on the upstream intron than on the downstream intron. To examine the involvement of the MATR3 bindings in the skipping of exon 2, we constructed a minigene, in which exon 2 and flanking intronic regions were inserted into the splicing reporter vector, pSPL3 (pSPL3-MATR3_ex2, Appendix A) [40]. As expected, the overexpression of MATR3-EGFP promoted the skipping of exon 2 in the transcripts from the minigene (Appendix A). We next artificially tethered MATR3 to the transcripts from pSPL3-MATR3_ex2 using the bacteriophage MS2 coat protein to simulate the endogenous binding of MATR3 to the upstream intron. We inserted 24xMS2 RNA hairpins into the pSPL3-MATR3 reporter minigene (pSPL3-MATR3_ex2+24xMS2) in the middle of the upstream intron (Figure 4C). We also made an effector construct expressing MATR3-EGFP tagged with the MS2 coat protein (MS2-MATR3-EGFP). RT-PCR analysis revealed that the tethering of MATR3-EGFP (MS2-MATR3-EGFP) promoted the skipping of exon 2 whereas MS2 alone or MS2-fused EGFP (MS2-EGFP) had no effect on the splicing of exon 2 (Figure 4C). MATR3-EGFP showed a limited effect on exon 2 skipping. These results suggest that the binding of MATR3 to the upstream intronic region represses exon 2 splicing.

### 3.4. The MATR3 mRNA Isoform Lacking Exon 2 Is Not Translated

We next investigated the translation status of *MATR3* mRNA lacking exon 2 (MATR3-Δex2) since 5UC-ASEs, including *MATR3* exon 2, contain translation initiation sites essential for translation. We analyzed the distribution of *MATR3* mRNAs on polyribosomes in cellulo using sucrose gradients. Since actively translated mRNA is associated with multiple ribosomes, it should be found in polysome fractions. In contrast, poorly translated mRNA, which is associated with only one or no ribosome, should be present in monosome fractions. Cytoplasmic extracts of HEK293 cells were fractionated on 15–45% sucrose gradients, and the exon 2 splicing in each fraction was assessed by RT-PCR. The analysis revealed that *MATR3* mRNA with exon 2 was highly concentrated in polysome fractions while MATR3-Δex2 mRNA was mainly distributed in monosome fractions (Figure 4D). In addition, Western blots with anti-MATR3 antibodies showed a minimal expression of truncated MATR3 proteins (Appendix A) despite the abundant expression of MATR3-Δex2 mRNA in these cells (Figure 4A). These results suggest that MATR3-Δex2 mRNA is not translated.

The low translation of MATR3-Δex2 mRNA led us to hypothesize that exon 2 is skipped to decrease MATR3 protein expression by converting *MATR3* mRNA to untranslatable MATR3-Δex2 mRNA since RBPs often form negative feedback loops in their expressions [46]. As expected, the overexpression of MATR3-EGFP decreased the expression of endogenous MATR3 protein in HEK293 cells (Figure 4E), although exon 2 skipping was promoted in these cells (Appendix A). These results suggest that MATR3 autoregulates its own protein levels through the production of MATR3-Δex2 mRNA.

## 4. Discussion

In this paper, we have identified a novel set of alternatively spliced internal exons, 5UC-ASEs. 5UC-ASEs have unique features, including the presence of a first methionine codon and large exon size. In addition, 5UC-ASEs are differentially spliced in the brain, muscle, and testis among 30 human tissues. In vivo and in silico analyses identified MATR3 as a specific regulator of 5UC-ASE splicing. Furthermore, evolutionary analysis revealed that 5UC-ASEs have emerged at least in cartilaginous fishes, have increased in amphibians, and are concentrated in genes involved in transcriptional regulation in mammals.

Human internal exons have a median length of 122 nt, and 5% of them exceed 300 nt [14,16]. We and others have previously reported that these large exons are enriched in C-nucleotides, are evolutionarily conserved, and are constitutively spliced [16,47]. The enriched C-nucleotides recruit a splicing factor, SRSF3, to achieve the constitutive splicing ubiquitously and also contribute to the formation of intrinsically disordered regions (IDRs), particularly in transcription factors, by increasing C-rich proline/serine codons [16]. In contrast, microexons, which are less than 30 nt in length, encode the interaction domains of neuronal factors and are alternatively spliced specifically in neuronal cells by the splicing factor, SRRM4 [15]. Although 5UC-ASEs have large exon sizes, they show no remarkable codon bias except for the methionine codon and are alternatively spliced in various tissues (Figure 1C and Figure 2A,B), suggesting that 5UC-ASEs form a distinct class of internal exons. In humans, terminal exons (first and last exons) are generally larger than internal exons [48,49]. A previous study suggested that large second exons, more than half of which contain translation initiation sites, have evolved from first exons [47]. This evolutionary mechanism is likely responsible for the large size of 5UC-ASEs.

We have shown that the alternative splicing of 5UC-ASEs is distinctly regulated in neuromuscular tissues (Figure 2A,B). MATR3 binds upstream of 5UC-ASEs to repress their splicing (Figure 3). Previous reports have shown that MATR3 is abundantly and poorly expressed in the brain and muscle, respectively, suggesting that MATR3 expression levels contribute to the specific alternative splicing regulation [22,23]. In addition, it has been well established that the coordination of alternative splicing networks promotes the development of various tissues including the brain and muscles [6]. Tissue-specific splicing factors such as NOVAs, RBFOXs, and PTBPs play a key role in alternative splicing regulation in neuromuscular tissues [6]. MATR3 is known to form splicing regulatory complexes with these factors [50,51]. MATR3 may regulate the alternative splicing of 5UC-ASEs by interacting with these splicing factors in addition to directly binding to the target RNA.

Due to the intrinsic link between alternative splicing and cell differentiation and homeostasis, the expression levels of splicing factors are tightly controlled [46]. Negative feedback mechanisms via RNA processing are often found to regulate their expression. For example, many splicing factors, including SR proteins and hnRNPs, autoregulate their expression by promoting the inclusion of alternative exons that introduce premature termination codons leading to the nonsense-mediated decay of mRNA [52]. In addition, U1A and TARDBP bind to the 3′ UTRs of their own mRNAs to promote RNA instability by impairing 3′ end processing and polyadenylation [53,54]. In the current paper, we have shown that MATR3 also autoregulates its expression through alternative splicing. MATR3 promotes the skipping of a 5UC-ASE, which switches the full-length *MATR3* mRNA for the canonical MATR3 protein to the untranslated *MATR3* mRNA isoform through the lack of a translation initiation site (Figure 4C,E and Appendix A). This demonstrates the unique function of a 5UC-ASE in regulating gene expression.

We found substantial expressions of the MATR3-Δex2 mRNA isoform in various tissues and cells (Figure 4A), suggesting that the feedback mechanism is broadly operational for fine-tuning MATR3 expression. In neuronal tissues, the *MATR3* mRNA isoform lacking both exon 2 (1089 nt) and exon 3 (62 nt) is highly expressed (Figure 4A), which is presumably not translated due to the absence of the translation initiation site in exon 2. A previous study reported the existence of the C-terminally truncated isoform in cytoplasmic processing bodies (P-bodies) in U2OS cells although the corresponding mRNA isoform was not identified [55]. The MATR3 protein undergoes degradation to reduce its expression level [56] and the cleavage of MATR3 protein by caspases generates a similar truncated MATR3 protein [57]. These observations imply that diverse MATR3 isoforms are generated in cells by both alternative splicing as well as post-translational processing to regulate MATR3 expression.

## 5. Conclusions

In the present paper, we have identified a unique set of skipping exons, 5UC-ASEs. 5UC-ASEs have evolved to play a role in transcriptional regulation, particularly in mammals. MATR3, the splicing regulator of 5UC-ASEs, has a 5UC-ASE and autoregulates its own protein expression by the alternative splicing-dependent deletion of a translation initiation site in *MATR3* mRNA. This regulatory mechanism may be broadly operational in the regulation of gene expression.

## Figures and Tables

**Figure 1 cells-14-00011-f001:**
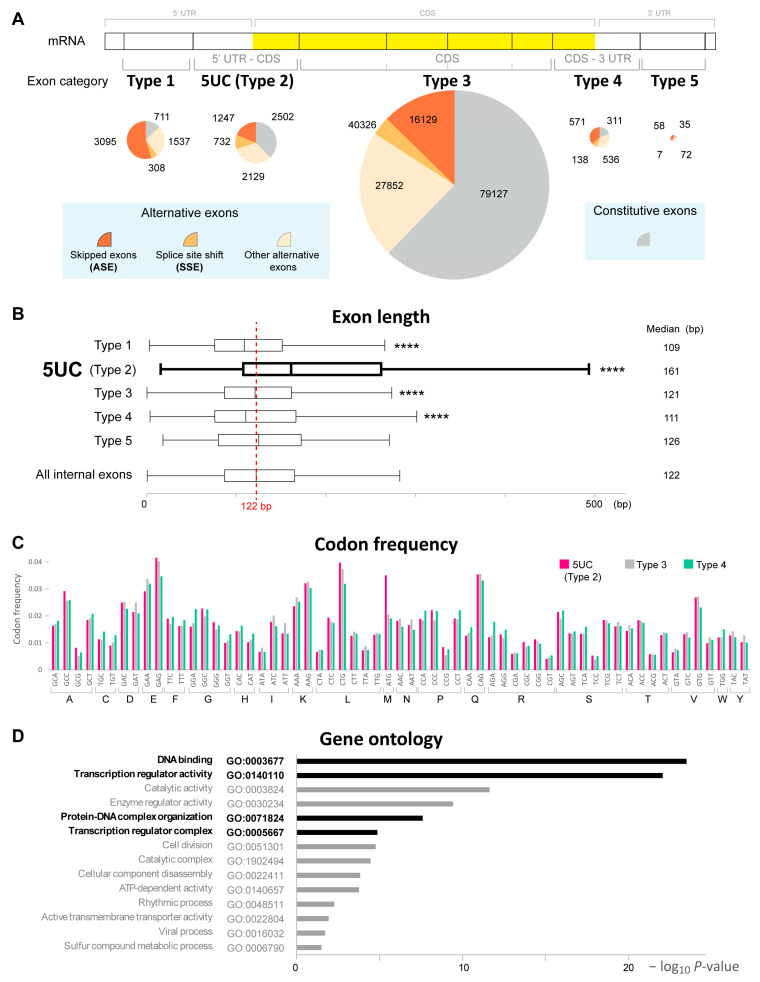
(**A**) Classification and number of internal exons in human protein-coding genes. The exons were divided into five categories based on the coding information, namely 5′UTR (Type 1), 5′UTR-CDS (5UC; Type 2), CDS (Type 3), CDS-3′UTR (Type 4), and 3′UTR (Type 5) (top panel). They were subdivided into constitutive exons and alternative exons including alternatively skipped exons (ASEs: cassette exons), splice-site-shifted exons (SSEs), and other alternative exons (bottom light-blue panels). Pie charts show the number of exons in each category. (**B**) Boxplot showing the exon lengths of Type 1, Type 2 (5UC), Type 3, Type 4, Type 5, and all internal exons. The median length of all internal exons was 122 nt (red dotted line). The plots show the interquartile range (boxes), the median (central red band), and the minimum and maximum except for the outliers at the ends of whiskers. **** *p* < 0.0001 as per Steel–Dwass test compared to all internal exons. (**C**) Bar graph showing the codon usage of Type 2, Type 3, and Type 4 exons. The *y*-axis indicates codon frequency, and the *x*-axis indicates triplet codons and corresponding amino acid codes. (**D**) Bar graph showing the significantly enriched (*p* < 0.05) gene ontology (GO) terms of Type 2 (5UC) genes. The GO terms associated with DNA binding and transcriptional regulation are indicated in black.

**Figure 2 cells-14-00011-f002:**
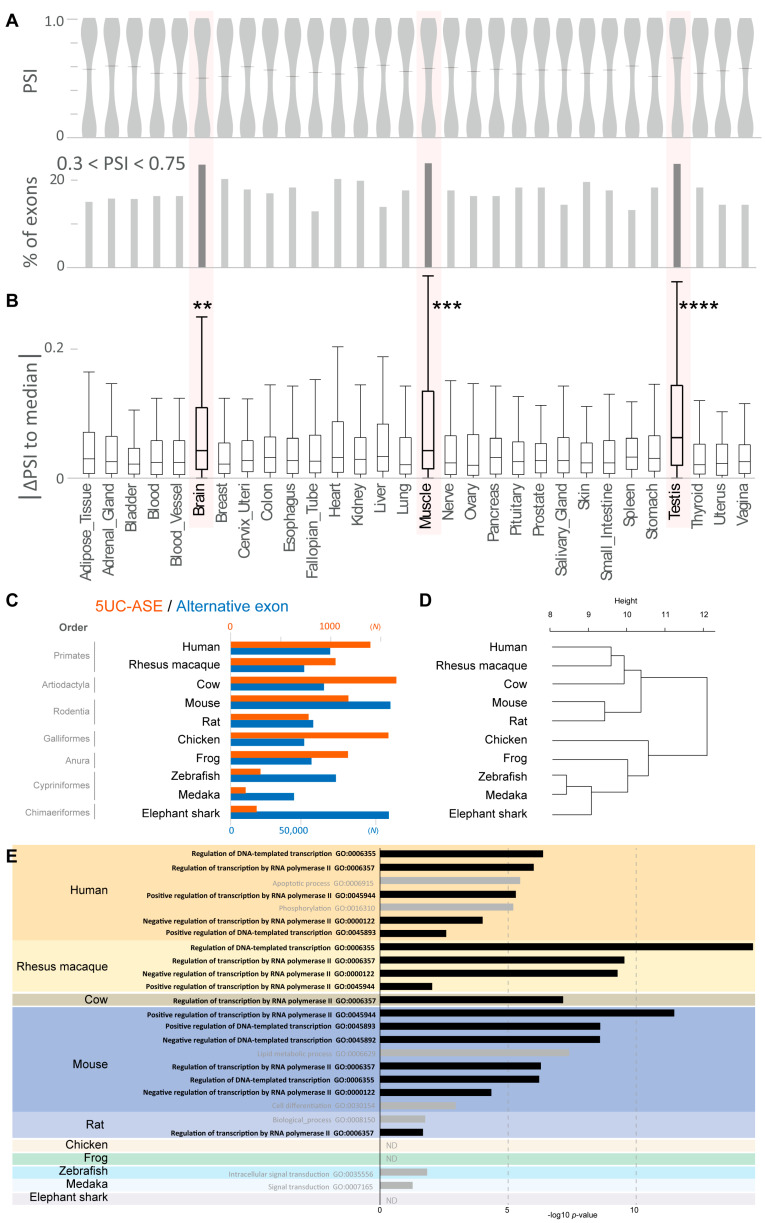
(**A**) Upper graph: Violin plot showing the percent spliced-in (PSI) values of 5UC-ASEs in 30 human tissues. Lower graph: Bar graph showing the ratios of moderately skipped 5UC-ASEs (0.3 < PSI < 0.75). (**B**) Boxplot showing the absolute values of tissue-ΔPSI [difference of the exon’s PSI in the tissue from the median of them in all tissues]. The plots show the interquartile range (boxes), the median (central red band), and the minimum and maximum except for the outliers at the ends of whiskers. ** *p* < 0.01, *** *p* < 0.001, and **** *p* < 0.0001 as per Steel–Dwass test. (**C**) Bar graph showing the numbers (*N*) of 5UC-ASEs (orange) and alternative exons (blue) in 10 vertebrate species including humans. (**D**) Dendrogram showing the orthologous relation of the genes containing 5UC-ASEs (5UC-ASE genes) in ten vertebrate species (**E**) Bar graph showing the *p*-values of GO terms associated with 5UC-ASE genes in each species. GO terms associated with at least 5% of the input genes were indicated. ND: not detectable. The GO terms associated with DNA binding and transcriptional regulation are indicated in black.

**Figure 3 cells-14-00011-f003:**
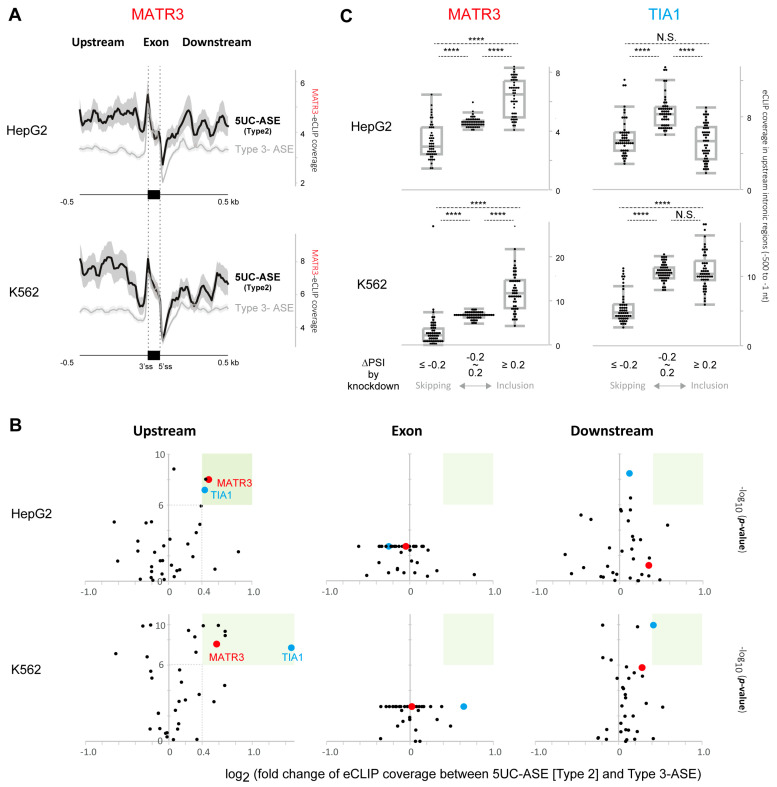
(**A**) Profile plots showing the distribution of MATR3–RNA interactions from 500 nt upstream to 500 nt downstream of 5UC-ASEs (Type 2) (black) and Type 3 ASEs (gray) in HepG2 and K562 cells. The standard error of the average coverage of MATR3-eCLIP reads is shown as a gray shade around the average curve. (**B**) Scatter plots showing the protein–RNA interaction coverage for 33 splicing factors in the upstream (−500 to −1 nt, upstream), exonic (Exon), and downstream (+1 to +500 nt, downstream) regions of 5UC-ASEs in HepG2 (upper graphs) and K562 cells (lower graphs). The *x*-axis indicates the log_2_ fold-change of eCLIP coverage between 5UC-ASE (Type 2) and Type 3 ASE and the *y*-axis indicates the −log_10_ *p*-value estimated by the exact Wilcoxon signed rank test. (**C**) Boxplot showing the relationship between the splicing of 5UC-ASEs and the recruitment of MATR3 (left) and TIA1 (right) to their upstream regions Median eCLIP coverage in each 50 nt bin of the upstream region (−500 to −1 nt) of 5UC-ASE is shown for the three 5UC-ASE groups divided based on ΔPSI by *MATR3*/*TIA1* silencing (ΔPSI ≥ 0.2, −0.2 < ΔPSI > 0.2, and ΔPSI ≤ −0.2). The plots show the interquartile range (boxes), the median (central red band), and the minimum and maximum except for the outliers at the ends of whiskers. **** *p* < 0.0001 as per Steel–Dwass test. N.S.: not significant.

**Figure 4 cells-14-00011-f004:**
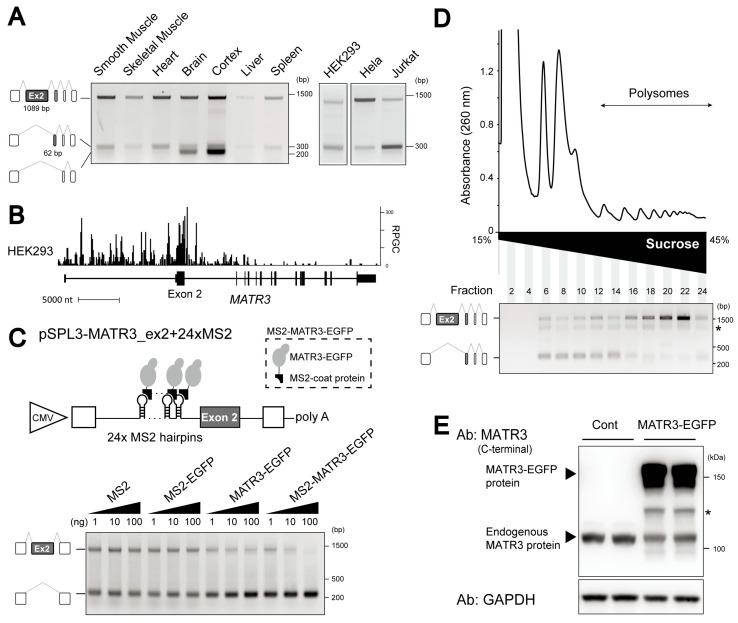
(**A**) RT-PCR showing alternative splicing of *MATR3* exon 2 in human tissues (left panel) and cell lines (right panel). In addition to exon 2, exon 3 is skipped in human brain and cortex, which was confirmed by sequencing analysis of the PCR products. (**B**) Distribution of MATR3 bindings on *MATR3* detected in our previously performed MATR3 CLIP-seq using HEK293 cells (DRR494386). (**C**) (Top panel) Schematic showing the reporter minigene, pSPL3-MATR3_ex2+24xMS2, carrying 24xMS2 hairpin loops in the middle of the upstream intron. Bottom panel: RT-PCR showing exon 2 splicing of the minigene. HEK293 cells were co-transfected with the reporter minigene and indicated quantities of effectors (MS2, MS2-EGFP, MATR3-EGFP, or MS2-MATR3-EGFP). (**D**) Distribution of *MATR3* mRNA isoforms with or without exon 2 in polysome fractions. Top panel: Cytoplasmic lysates of HEK293 cells were separated on a 15–45% sucrose gradient. Absorbance at 260 nm is shown on the *y* axis. Fractions from the top of the gradient to the bottom are shown from left to right on the *x*-axis. Fractions were collected in 24 equal volumes, and RNA was extracted from each fraction. Bottom panel: RT-PCR analysis of exon 2 splicing at the indicated fractions. The asterisk denotes the heteroduplex bands of upper and lower PCR products. (**E**) Overexpression of MATR3-EGFP downregulates endogenous MATR3 protein expression. HEK293 cells were transfected with MATR3-EGFP or control (EGFP alone: Cont) expression vectors. After 72 h, total cell lysates were harvested, and Western blotting analysis was performed with indicated antibodies (Abs). The asterisk denotes fragmented MATR3-EGFP protein that was also detected in the blot with anti-GFP antibody (Appendix A).

**Table 1 cells-14-00011-t001:** Classification and number of internal exons in human protein-coding genes.

Exon Class	Total	Alternatively Skipped Exon (ASE)	Splice-Site-Shifted Exons (SSE)	Other Alternative Exon	Constitutive Exon
Type 1	5651	3095	308	1537	711
5UC (Type 2)	6610	1247	732	2129	2502
Type 3	127,144	16,129	4036	27,852	79,127
Type 4	1556	571	138	536	311
Type 5	172	58	7	72	35

**Table 2 cells-14-00011-t002:** Numbers of genes, transcript isoforms, and exons of protein-coding and 5UC-ASE genes in ten vertebrate species.

	Protein-Coding Genes				5UC-ASE Genes		
Species	Genes	Transcript Isoforms	Exons	Alternative Exons	Genes	Transcript Isoforms	ASEs
Human	20,042	64,128	289,850	66,286	1130	3153	1398
Rhesus macaque	21,369	48,237	256,946	48,971	878	1581	1048
Cow	23,492	59,313	268,947	62,243	1388	2065	1657
Mouse	21,833	102,034	451,167	106,475	907	2480	1178
Rat	23,049	45,829	264,477	54,998	701	899	778
Chicken	16,711	44,314	241,811	49,013	1293	1865	1578
Frog	21,759	48,976	268,133	53,862	944	1701	1173
Zebrafish	25,107	50,807	313,568	70,100	268	407	298
Medaka	23,587	37,433	253,363	42,102	168	238	181
Elephant shark	19,415	49,321	260,786	105,543	243	359	259

## Data Availability

The data presented in this study are available upon request from the corresponding author.

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
