# Peer review of "Evolutionarily Developed Alternatively Spliced Exons Containing Translation Initiation Sites"

_cells, 2024, doi:10.3390/cells14010011_

Round 1
Reviewer 1 Report
Comments and Suggestions for Authors
General comments:
This manuscript by Takeda et al. reports important features of various alternatively spliced exons and advances current knowledge. The manuscript is well written, apart from some 'over-use' of acronyms, particularly in the abstract. This reviewer suggests removing all acronyms, including those not defined or explained (eg CDS, PSI) from the abstract and introduction, and simplifying explanations so that the abstract is comprehensible to the interested and not-necessarily-expert reader. Fundamentally important data are discussed; clearer summarizing of the key findings in the abstract would be helpful.
Specific comments:
While the intent of the authors to generate meaningful terms for the different exon categories understood, this reviewer found the terms too similar (eg 5U-Es and 5UC-Es), confusing and not entirely intuitive, further confounded by the extensive use of acronyms for so many other terms. Would the authors consider simple 'Type 1, Type 2 etc categories, or alternatively, terms that are 'more different'? Also, writing most terms in full, eg 'alternative splicing', 'alternatively spliced exons' etc will enhance readability and appreciation of this very nice work. A table cited in the introduction and showing the exon category, name, characteristics (size range etc), references/datasource might be helpful.
Line 64: ALS and FTLD are only mentioned once- write in full, no need for abbreviation.
Line 86-91- rephrase this section to improve clarity. 'Based on the genomic locations of internal exons in transcript isoforms, we defined a cluster of overlapping exons having only identical 5’ and 3’ splice sites as a constitutively spliced exon. A unique exon without any overlapping transcripts was also defined as a constitutively spliced exon. The other exons were classified as AEs. The AEs were further subdivided into cassette exons (skipping exons, SEs), AEs with alternative 5’ or 3’ splice sites (splice site shifted exons, SSs), and the other AEs.
Line 280: Please clarify- 'The large exon sizes for 5UC-Es are similarly observed between constitutively spliced exons and AEs (Figure S1A).' I don't believe this refers to all alternative exons?
Line 291: rephrase- 'We noticed that 5UC-Es contain a substantial number of SEs' - the 5UC exons cannot contain exons. eg We observed that a substantial number of the 5UC-Es were alternatively spliced (skipped, 5UC-SEs, 1,247 exons); or alternatively, 'the 5UC-SEs category included a substantial number of skipped exons'.
Line 339, Tabel 2: For clarity, a delineation between the columns related to Protein-coding gene and the 5UC-SE gene sections is recommended.
Author Response
Comments 1: General comments: This manuscript by Takeda et al. reports important features of various alternatively spliced exons and advances current knowledge. The manuscript is well written, apart from some 'over-use' of acronyms, particularly in the abstract. This reviewer suggests removing all acronyms, including those not defined or explained (eg CDS, PSI) from the abstract and introduction, and simplifying explanations so that the abstract is comprehensible to the interested and not-necessarily-expert reader. Fundamentally important data are discussed; clearer summarizing of the key findings in the abstract would be helpful.
Response 1: Thank you for pointing this out. As suggested, we have revised the abstract and introduction to minimize acronyms and improve clarity for better understanding. Since “5UC-ASE” is an essential term for our manuscript, we hope to keep it in the abstract and introduction.
Comments 2: Specific comments: While the intent of the authors to generate meaningful terms for the different exon categories understood, this reviewer found the terms too similar (eg 5U-Es and 5UC-Es), confusing and not entirely intuitive, further confounded by the extensive use of acronyms for so many other terms. Would the authors consider simple 'Type 1, Type 2 etc categories, or alternatively, terms that are 'more different'? Also, writing most terms in full, eg 'alternative splicing', 'alternatively spliced exons' etc will enhance readability and appreciation of this very nice work. A table cited in the introduction and showing the exon category, name, characteristics (size range etc), references/datasource might be helpful.
Response 2: We apologize for the use of confusing terms. According to the reviewer’s suggestions, we have replaced “5U”, “5UC,” “C”, “C3U”, and “3U” with “Type 1”, “Type 2”, “Type 3”, “Type 4”, and “Type 5”, respectively (Figure 1A). After specifying the type 1 to 5 exon categories, we have used the specific term “5UC” for the type 2 exons, considering their importance in this paper. Since “SEs” (skipped exons) and “Es” (exons) are also confusing, we have replaced “SEs” with “ASEs” (alternatively skipped exons) and stopped using “Es” to improve readability. We have also written “alternative splicing” and “alternative exons” in full throughout the manuscript.
Comments 3: Line 64: ALS and FTLD are only mentioned once- write in full, no need for abbreviation.
Response 3: Thank you for pointing these out. We have corrected them.
Comments 4: Line 86-91- rephrase this section to improve clarity. 'Based on the genomic locations of internal exons in transcript isoforms, we defined a cluster of overlapping exons having only identical 5’ and 3’ splice sites as a constitutively spliced exon. A unique exon without any overlapping transcripts was also defined as a constitutively spliced exon. The other exons were classified as AEs. The AEs were further subdivided into cassette exons (skipping exons, SEs), AEs with alternative 5’ or 3’ splice sites (splice site shifted exons, SSs), and the other AEs.
Response 4: We apologize for the confusing description. We have changed the sentence to “A unique internal exon with no other overlapping transcript isoforms was defined as a constitutively spliced exon. When multiple transcript isoforms with internal exons are generated from a gene, an internal exon present in all transcript isoforms was also defined as a constitutively spliced exon. The other exons were classified as alternatively spliced exons. The alternatively spliced exons were further subdivided into cassette exons (alternatively skipped exons, ASEs), exons with alternative 5’ or 3’ splice sites (splice site shifted exons, SSEs), and the other alternatively spliced exons.”
Comments 5: Line 280: Please clarify- 'The large exon sizes for 5UC-Es are similarly observed between constitutively spliced exons and AEs (Figure S1A).' I don't believe this refers to all alternative exons?
Response 5: Thank you for pointing this out. We intended to explain constitutively and alternatively spliced 5UC exons. We have corrected the sentence as follows. “The large exon sizes are similarly observed between constitutively spliced and alternatively spliced 5UC exons (Figure S1A)”.
Comments 6: Line 291: rephrase- 'We noticed that 5UC-Es contain a substantial number of SEs' - the 5UC exons cannot contain exons. eg We observed that a substantial number of the 5UC-Es were alternatively spliced (skipped, 5UC-SEs, 1,247 exons); or alternatively, 'the 5UC-SEs category included a substantial number of skipped exons'.
Response 6: Thank you for the suggestion. We have revised the sentence as indicated. “We observed that a substantial number of the 5UC exons were alternatively skipped (5UC-ASEs, 1,247 exons),”.
Comments 7: Line 339, Tabel 2: For clarity, a delineation between the columns related to Protein-coding gene and the 5UC-SE gene sections is recommended.
Response 7: Thank you for pointing this out. We have modified the table as suggested.
Reviewer 2 Report
Comments and Suggestions for Authors
Alternative splicing (AS) is a critical process in the generation of various protein isoforms, that influence cell differentiation and tissue development. In addition to internal coding exons, AS affects the exons with translation initiation codons; however, little is known about these exons.
In this manuscript, the authors performed a systematic classification of human alternatively spliced exons (AEs) using CDS information. The results showed that about 10% of AEs contain translation initiation codons [AEs harboring 5'UTR-CDS (5UC-SEs)]. Evolutionary analysis revealed that 5UC-SEs have already appeared in cartilaginous fish and are concentrated in the genes involved in transcriptional regulation in mammals. In humans, 5UC-SEs are alternatively spliced, particularly in the brain, muscle, and testis. In order to explore the splicing mechanism of 5UC-SEs, the authors analyzed eCLIP datasets of 33 splicing factors and found that MATR3 binds to the upstream introns of 5UC-SEs to repress their splicing. Basing cell line and tissue, the results indicated that MATR3 binds upstream to its own exon 2, a 5UC-SE, repressing its splicing and reducing MATR3 translation to autoregulate its expression level. This regulatory mechanism may be broadly operational in the regulation of gene expression.
There are some minor issues to address:
1. For exons that can be spliced out before mature mRNA formation, theoretically, they should be multiples of three to avoid frameshift mutations, resulting in highly homologous proteins. In MATR3, exon 2 is a multiple of three. The authors suggest that exon 3 could also be skipped; it is recommended to provide a diagram showing the length data of each exon in MATR3 to confirm if it is a multiple of three. If not, why?
- There are many variants of MATR3 mRNA in databases. Can the bands from RT-PCR alone provide accurate information about the true nature of the bands? Expression of variant forms should vary greatly across different tissues; why hasn't sequencing validation been conducted?
- In Figure 4D, are the data derived from cell lines or tissues?
- After deleting exon 2 in MATR3, approximately 363 amino acids are missing. Why is this variant not reflected in the Western blot (Figure S3D and other WB figures)?
- The notation for MATR3's minigene is somewhat inconsistent in the text. In Figure 4, it is written as pSPL3-MATR3_MS2, while in Figure S3, it is pSPL3-MATR3_ex2. The latter notation appears more appropriate.
- In Figure 4E, regarding the endogenous expression of MATR3, are there full-length and exon 2-deleted bands? The bands here is due to the antibody? The N-terminal or C-terminal anti-MATR3 antibody should be specified in the figure. For proteins fused with EGFP, why not consider using an anti-EGFP antibody?
- In the electrophoresis graph of Figure 4D, what is the cause of the unclear bands that appear? Are they due to the deletion of exon 3?
- Table S4, which contains primer sequences, is very important and should be included in the main text rather than as a supplementary.
Author Response
Comments 1: For exons that can be spliced out before mature mRNA formation, theoretically, they should be multiples of three to avoid frameshift mutations, resulting in highly homologous proteins. In MATR3, exon 2 is a multiple of three. The authors suggest that exon 3 could also be skipped; it is recommended to provide a diagram showing the length data of each exon in MATR3 to confirm if it is a multiple of three. If not, why?
Response 1: As the reviewer sharply pointed out, MATR3 exon 3 is 62 bp, which is not a multiple of three. Since exon 3 is skipped together with exon 2 (Figure 4A, brain and cortex), the mRNA isoform lacking exon 3 (+ exon 2) has no translation initiation site, leading to its little translation as indicated for the MATR3-Δex2 mRNA isoform in the manuscript. This is probably the reason why the exon 3 skipping was abundantly observed (Figure 4A), although the skipping theoretically causes a frameshift. We have indicated the size of exons 2 and 3 in Figure 4A and described the existence of the untranslated MATR3 mRNA isoform lacking these exons in the “Discussion” section.
Comments 2: There are many variants of MATR3 mRNA in databases. Can the bands from RT-PCR alone provide accurate information about the true nature of the bands? Expression of variant forms should vary greatly across different tissues; why hasn't sequencing validation been conducted?
Response 2: We sequenced the RT-PCR fragments in Figure 4A and confirmed the skipping of exon 2 and the skipping of exons 2 and 3 in the corresponding fragments. We have newly indicated the sequencing results as Figure S3A.
Comments 3: In Figure 4D, are the data derived from cell lines or tissues?
Response 3: We consistently used HEK293 cells in the experiments shown in Figure 4BCDE. We have explicitly mentioned the use of HEK293 cells in the “Materials and Methods” (polysome profiling analysis [section 2.7.]) and in the legend of Figure 4D.
Comments 4: After deleting exon 2 in MATR3, approximately 363 amino acids are missing. Why is this variant not reflected in the Western blot (Figure S3D [currently S3E] and other WB figures)?
Response 4: We apologize for the inadequate explanation of our experiments on the translation of the MATR3 mRNA isoform lacking exon 2 (MATR3-Δex2). Translation of MATR3-Δex2 mRNA should be impaired due to the absence of the canonical translation initiation site. To analyze the actual translation status of MATR3-Δex2 mRNA, we performed a polysome profiling analysis in Figure 4D. This technique separates translated mRNAs on a sucrose gradient according to the number of associated ribosomes. Actively translated mRNA is present in polysome fraction because it is associated with multiple ribosomes. In contrast, poorly translated mRNA is associated with only one or no ribosome and is present in monosome fraction. Our experiment showed that MATR3-Δex2 mRNA was mostly distributed in monosome fractions, whereas MATR3 mRNA with exon 2 was highly concentrated in polysome fractions (Figure 4D). This suggests the minimal translation of MATR3-Δex2 mRNA. Therefore, the protein expression of MATR3-Δex2 mRNA was not detected in our Western blotting analyses. We have added the description of the polysome profiling experiment to the Results section (section 3.4.) for a better understanding of our analysis.
Comments 5: The annotation for MATR3's minigene is somewhat inconsistent in the text. In Figure 4, it is written as pSPL3-MATR3_MS2, while in Figure S3, it is pSPL3-MATR3_ex2. The latter notation appears more appropriate.
Response 5: We apologize for the confusing naming of our minigenes. We have changed “pSPL3-MATR3_MS2” to “pSPL3-MATR3_ex2+24xMS2” to clarify the insertion of 24xMS2 hairpins into the pSPL3-MATR3_ex2 minigene.
Comments 6: In Figure 4E, regarding the endogenous expression of MATR3, are there full-length and exon 2-deleted bands? The bands here is due to the antibody? The N-terminal or C-terminal anti-MATR3 antibody should be specified in the figure. For proteins fused with EGFP, why not consider using an anti-EGFP antibody?
Response 6: Thank you for pointing this out. We performed a Western blot with anti-GFP antibody (Figure S3F) and found that the band between MATR3-EGFP (~150 kDa) and endogenous MATR3 (~110 kDa) in Figure 4E is a fragment of MATR3-EGFP protein, as the same size bands are observed in the anti-GFP blot (Figure S3F, asterisk). As described in Response 4, the MATR3 mRNA lacking exon2 is barely translated. Therefore, the corresponding protein band was not obvious in the Western blots (Figures 4E and S3E). In Figure 4E, anti-MATR3 antibody (C-terminal) was used to show the expression of the endogenous MATR3 protein in addition to the overexpressed MATRE3-EGFP protein. According to the reviewer’s comments, we have indicated the use of the C-terminal antibody in Figure 4E.
Comments 7: In the electrophoresis graph of Figure 4D, what is the cause of the unclear bands that appear? Are they due to the deletion of exon 3?
Response 7: Thank you for pointing this out. In Figures 4AC and S3BD, where the unclear band is not visible, PCRs were performed with ~30 cycles. We noticed that the unclear band appeared at 32 cycles and became obvious at 35 cycles (bottom figure), suggesting that extensive PCR amplification causes the formation of the band. In Figure 4D, where the unclear band is present, PCR was performed with 35 cycles to detect the bands for MATR3 mRNA due to the small amounts of RNA in each fraction. Sanger sequencing analysis of the PCR products detected only the sequences corresponding to the upper band (the mRNA isoform with exon 2) or the lower band (MATR3-Δex2 mRNA), suggesting that the unclear band is a heteroduplex of the upper and lower PCR products. We have described the existence of the heteroduplex band in the legend of Figure 4D.

Comments 8: Table S4, which contains primer sequences, is very important and should be included in the main text rather than as a supplementary.
Response 8: Thank you for the suggestion. We have moved the primer sequences to the “Materials and Methods” (RNA extraction and RT-PCR [section 2.5.]) .
Round 2
Reviewer 2 Report
Comments and Suggestions for Authors
There are no further concerns to address.